# B-STaR: Monitoring and Balancing Exploration and Exploitation in Self-Taught Reasoners

**Weihao Zeng**[*1]    **Yuzhen Huang**[*1]    **Lulu Zhao**[2]    **Yijun Wang**[3]    **Zifei Shan**[3]    **Junxian He**[1]
[1]The Hong Kong University of Science and Technology    [2]BAAI    [3]Tencent
{wzengak,yhuanghj,junxianh}@cse.ust.hk

## Abstract

In the absence of extensive human-annotated data for complex reasoning tasks, self-improvement – where models are trained on their own outputs – has emerged as a primary method for enhancing performance. Recently, the approach to self-improvement has shifted toward a more dynamic, online fashion through iterative training. However, the critical factors underlying the mechanism of these iterative self-improving methods remain poorly understood, such as under what conditions self-improvement is effective, and what are the bottlenecks in the current iterations. In this work, we identify and propose methods to monitor two pivotal factors in this iterative process: (1) the model's ability to generate sufficiently diverse responses (*exploration*); and (2) the effectiveness of external rewards in distinguishing high-quality candidates from lower-quality ones (*exploitation*). These factors are inherently dynamic throughout the iterative process, yet prior research rarely discusses their evolution – leaving unclear why models often stagnate after only a few iterations. Using mathematical reasoning as a case study, we begin with a quantitative analysis to track the dynamics of exploration and exploitation, discovering that a model's exploratory capabilities rapidly deteriorate over iterations, and the effectiveness of exploiting external rewards diminishes as well. Motivated by these findings, we introduce B-STaR, a **S**elf-**Ta**ught **R**easoning framework that autonomously adjusts configurations across iterations to **B**alance exploration and exploitation, thereby optimizing the self-improving effectiveness based on the current policy model and available rewards. Our experiments on mathematical reasoning, coding, and commonsense reasoning demonstrate that B-STaR not only enhances the model's exploratory capabilities throughout training but also achieves a more effective balance between exploration and exploitation, leading to superior performance. Crucially, this work deconstructs the opaque nature of self-training algorithms, providing interpretable insights into their dynamics and highlighting current limitations to guide future research.[1]

## 1 Introduction

Large language models possess advanced reasoning capabilities such as mathematical problem-solving (Cobbe et al., 2021), coding challenges (Chen et al., 2021) or commonsense reasoning (Clark et al., 2018). However, the challenge of acquiring extensive, high-quality human-curated datasets remains a significant barrier to further enhancing these reasoning abilities. As tasks grow in complexity, the reliance on human-generated data becomes increasingly unsustainable, necessitating alternative approaches to training.

To tackle this issue, methods rooted in the concept of "self-improvement" (Huang et al., 2022), such as STaR (Zelikman et al., 2022), RFT (Yuan et al., 2023), and ReST (Gulcehre et al., 2023; Singh et al., 2023), provide more cost-effective and scalable solutions. Self-improvement follows an iterative process where the model generates responses, from which the better are selected to create

---

[*]Equal Contribution.
[1]We open-source our code at https://github.com/hkust-nlp/B-STaR.

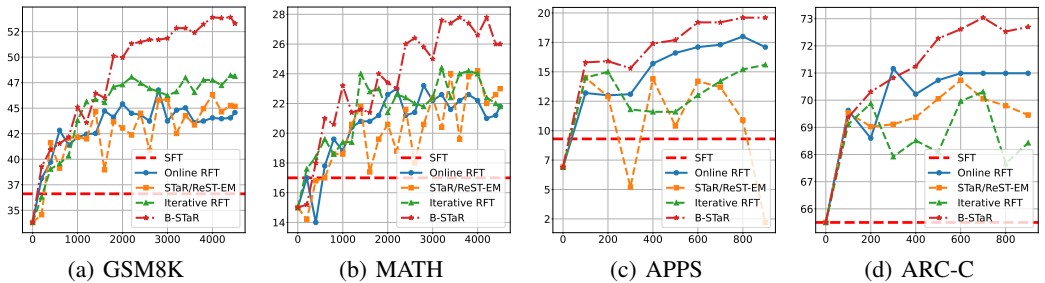

Figure 1: Pass@1 accuracy over training steps on GSM8K, MATH, APPS, and ARC-Challenge. The GSM8K, MATH, and ARC-C results are from Mistral-7B, while the APPS scores are obtained based on Llama3-8B. We compare multiple baselines, including SFT, Online RFT, STaR/ReST-EM, and Iterative RFT, against our proposed B-STAR. For further details on the experimental settings, please refer to §4.1.

higher-quality data for further refinement (Hosseini et al., 2024). This continuous loop allows the model to improve its performance over time, reducing the need for large amounts of human-generated data. Ultimately, this approach enhances the model's ability to handle complex reasoning tasks, pushing the limits of its capabilities (Havrilla et al., 2024).

Despite significant advancements, we still lack a deep understanding of the key factors that drive successful self-improvement and the internal optimization mechanisms remain largely opaque. Understanding the critical components and bottlenecks of these self-improving methods is particularly important given that performance of current self-improving approaches does not scale well with increased compute and saturates very quickly after merely 3 to 5 iterations (Singh et al., 2023; Wu et al., 2024). In this paper, we address the following questions: (1) What key factors play a decisive role in the self-improvement process? (2) How can these factors be used to analyze the limitations of current self-improvement methods from a unified perspective? and (3) How can we leverage these factors to guide the self-improvement process, ultimately maximizing performance gains?

To this end, we identify two crucial capabilities of the model during its self-improvement process: (1) Exploration – the model's ability to generate correct and diverse responses among multiple generated candidates (Singh et al., 2023; Wang et al., 2024a), and (2) Exploitation – the effectiveness of external rewards (e.g., a reward model or final answer supervision) in selecting high-quality solutions from these candidates (Wang et al., 2024b; Sun et al., 2024). Intuitively, these two factors are dynamic, evolving throughout the training process, a phenomenon that remains underexplored despite its critical importance. In this work, we first conduct empirical analysis to quantitatively monitor the dynamics of exploration and exploitation during iterative training processes. We observe that both capabilities may stagnate or even decline, and imbalances between them can hinder the model's ongoing improvement.

Motivated by these insights, we propose a novel approach for self-improvement that automatically monitors and balances these dynamic factors to optimize the use of the current policy and reward. This involves adjusting configurations that influence exploration and exploitation, such as sampling temperature and reward thresholds. These configurations are adaptively modified throughout training in terms of our proposed metric, *balance score*. This new metric assesses the potential of a query based on the current model's exploration and exploitation capabilities, and our method automatically balances exploration and exploitation behaviors to maximize the average balance scores. We refer to this method as B-STAR, a Balanced Self-Taught Reasoner.

The experimental results from mathematical problem-solving, coding challenges and commonsense reasoning demonstrate that B-STAR significantly surpasses other self-improvement methods through balanced exploration and exploitation. For instance, B-STAR achieves a significant improvement in Pass@1 on both GSM8K and MATH, surpassing various self-improving variants while maintaining a steady upward trajectory, as depicted in Figure 1. Furthermore, we demonstrate that exploration-related metrics, such as Pass@32, are continuously improving without any notable degradation.

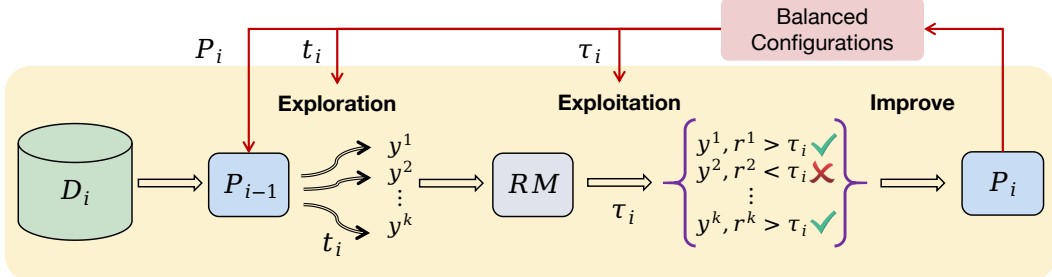

Figure 2: Illustration of the B-STAR approach. In each iteration, we first identify the configurations – temperature $t_i$ and reward threshold $\tau_i$ – that maximize the average balance scores using a small subset of training queries. Next, we apply the optimal temperature and threshold to generate and reward the full set of training queries. Finally, we update the model based on the selected data.

## 2 MONITORING EXPLORATION AND EXPLOITATION IN SELF-IMPROVEMENT

### 2.1 BACKGROUND: SELF-IMPROVEMENT

Given a pre-trained model $P_0$ and a training set $D = (x_i, y_i)_{i=1}^{N}$, where $x_i$ denotes the training query and $y_i$ represents the response, the goal of self-improvement is to iteratively generate high-quality responses from the current model and update the model with these self-generated data. Let $T$ represent the total number of iterations, with the model at the start of the $t$-th iteration denoted as $P_{t-1}$. In the first iteration, $P_0$ is typically fine-tuned on the initial dataset $D$, then each subsequent iteration involves three critical steps generally (Yuan et al., 2023; Gulcehre et al., 2023):

(1) Generating (Sampling): For each query $x_i$, the model $P_{t-1}$ generates $K$ candidate responses, forming a new, self-generated dataset.

(2) Rewarding (Verifying): A reward function $r(x, y)$ is applied to score and select the high-quality responses from the self-generated dataset. This reward can be binary and utilize additional supervision, for example, in problem-solving tasks in the math and code domains, it is common to use the final answer matching or the result of passing unit tests as binary feedback (Yuan et al., 2023; Chen et al., 2022). In a more sophisticated case, $r(x, y)$ can be parameterized by a reward model outputting continuous scores, using outcome-based reward models (ORMs) (Li et al., 2022) or process-based reward models (PRMs) (Uesato et al., 2022; Lightman et al., 2023; Havrilla et al., 2024).

(3) Improving: The selected dataset is used to update $P_{t-1}$, producing $P_t$. To differentiate the generation model $M$ from the reward model, $M$ is also referred to as the policy model following RL literature. In reasoning tasks that we are going to focus on, SFT loss is commonly used in the Improving step due to its robustness and scalability (Pang et al., 2024; Dubey et al., 2024), as more sophisticated RL losses can be unstable to optimize and scale up. When SFT loss is adopted, the Rewarding step aims to reject low-quality responses and use remaining high-quality data for training, thus this process is also referred to as rejection fine-tuning (RFT, Yuan et al. (2023)). In Appendix C, we provide detailed explanations of self-improvement, including reward functions and RFT.

**Discussion on Online Learning.** The iterative procedure described above can be contextualized within the reinforcement learning framework (Singh et al., 2023), and the iterative design shifts the vanilla offline training towards a more dynamic online variant – when iteration intervals are short and the optimizer is inherited between iterations, the training essentially transforms into a fully online learning algorithm. Iterative RFT implementations in previous works typically adopt long iteration intervals where each iteration processes all available queries (Zelikman et al., 2022; Sun et al., 2024). Sometimes, these implementations opted to restart the training from the initial checkpoint rather than from the last saved model (Zelikman et al., 2022; Singh et al., 2023). However, we argue that always starting from the beginning is not scalable for large datasets, particularly in a streaming setting, instead, a continuous training approach more aligned with RL principles is preferable. Transitioning from offline to online training, online RFT (Shao et al., 2024) has demonstrated its superiority over traditional offline RFT methods by switching iterations more frequently, ensuring that the synthetic data remains on-policy. In this study, we explore online RFT as our primary framework.

## 2.2 THE CRITICAL FACTORS – EXPLORATION AND EXPLOITATION

To maximize the gains from self-improvement training and fundamentally enhance the model's ability using its own outputs, the key is to achieve scalable self-improvement training, where the model's performance scales with increased computational investment in the training algorithm. However, all previous works show quick saturation after merely 3-5 self-improvement iterations (Singh et al., 2023; Wu et al., 2024), hypothesizing that the model's own outputs can only lead to limited gains. In this work, we seek to dive deeper into the currently opaque process of self-improvement, to understand the critical factors that determine whether self-improvement succeeds or fails.

Intuitively, for a certain iteration of training, we argue that two high-level conditions must be met for the model to make progresses: (1) *Diverse Exploration for High-Quality Responses:* When multiple candidates are sampled from the model, a portion of them must be high-quality responses. This is particularly important for queries where the model fails to produce satisfactory outputs using greedy decoding. Such diversity enables the discovery of responses that cannot be reached through greedy decoding. (2) *Effective Reward Function Discrimination:* The reward function $r(x, y)$ must reliably distinguish high-quality candidates from lower-quality ones. We refer to the two conditions as *exploration* and *exploitation* respectively, and we provide their analogies to RL in Appendix G. If either of these conditions is unmet – such as when the model produces responses overly similar (i.e. lack of diversity), or when the reward function fails to identify high-quality responses – the self-improvement will be limited on the gains.

**Exploration and Exploitation are Moving Targets.** Both exploration and exploitation are dynamically influenced by the policy model during the self-improvement process. On one hand, after multiple iterations, the policy model may overfit the task, failing to explore diverse responses and instead generating highly similar outputs (i.e., a lack of exploration). Training on these highly similar responses is unlikely to yield significant improvements. On the other hand, if the model generates excessively diverse responses, resulting in a distribution that deviates significantly from the training data distribution of the reward model, it becomes challenging for the reward model to reliably distinguish high-quality responses (i.e., a lack of exploitation). Thus, maintaining a dynamic balance between exploration and exploitation is essential throughout the self-improving process. However, such dynamics are rarely discussed in prior research, Next, we propose methods to quantify exploration and exploitation, enabling us to monitor their dynamics during training and deepen our understanding of the mechanisms underlying self-improvement.

**Quantifying Exploration and Exploitation.** In this work, we mainly focus on complex problem-solving tasks such as math and coding domains, where the correctness of the responses can be easily verified on labeled datasets.[2] This property facilitates the quantification of exploration and exploitation, for which we detail the metrics below:

- *Exploration:* Pass@K, which measures whether there is at least one correct response during K sampled candidates, is a straightforward metric for assessing exploration, as it directly reflects whether the model is able to explore correct solutions. However, Pass@K can be noisy, as it only counts a single correct response, while it is desirable to assess whether the model can explore multiple correct response as well. To this end, we propose to track Pass@K-S as well, which measures whether there are at least S unique correct responses among K sampled candidates. Pass@K-S serves as a more stable proxy to exploration than Pass@K. Pass@K is essentially Pass@K-1 following such a definition. Besides Pass@K and Pass@K-S, we also track diversity of the generations using Distinct Equations proposed by Wu et al. (2024), which measures the proportion of unique equations among all correct generated responses.

- *Exploitation:* Best-of-K accuracy measures whether the top one response ranked by the reward function is correct, directly reflecting how well the reward can potentially select a good response. Since it is typically required to select multiple responses rather than one in self-improving training, we are interested in the reward's exploitation to select multiple responses as well. To this end, we come up with the Reward@K-S metric, which measures whether the top S candidates ranked by the reward are all correct or not. Best-of-K accuracy is a special case of Reward@K-S when S

---

[2]Strictly speaking, the response may contain incorrect steps even though the final answer is correct, we do not further distinguish this difference following others (Zelikman et al., 2022) as it is not the focus of this work.

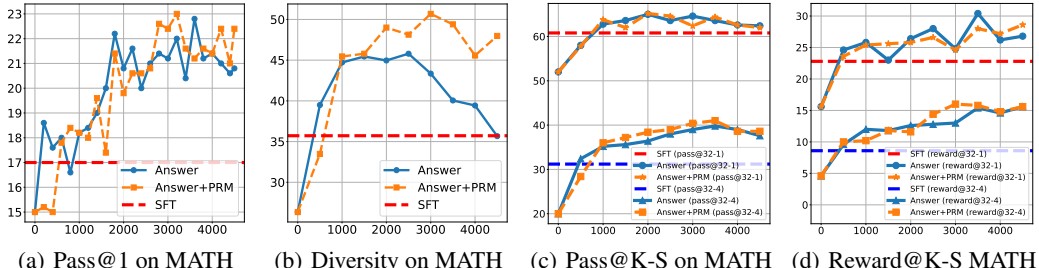

(a) Pass@1 on MATH    (b) Diversity on MATH    (c) Pass@K-S on MATH    (d) Reward@K-S MATH

Figure 3: Pass@1, Diversity, Pass@K-S and Reward@K-S over training steps on MATH. SFT refers to direct fine-tuning using the original dataset, "Answer" indicates matching against the ground-truth final answer, and "Answer + PRM" combines ground-truth final answer matching with PRM reward.

is equal to 1. One might think that Reward@K-S would be equivalent to Pass@K-S if only the final answer is used to select responses, potentially making Reward@K-S less useful in such cases. However, we emphasize that the exploitation metrics are mainly used to measure the effectiveness when additional reward models are integrated, as we will show next in §2.3 that our reward function combines final answer supervision and a reward model.

Next, we conduct a case study to dive into self-improving training through tracking these metrics.

## 2.3 DYNAMICS OF EXPLORATION AND EXPLOITATION – A CASE STUDY IN MATHEMATICAL PROBLEM SOLVING

In this section, we perform a case study to analyze the dynamics of exploration and exploitation in a mathematical reasoning task. Specifically, we follow Singh et al. (2023) to adopt MATH (Hendrycks et al., 2021) training set as the training data, evaluating on the test split of MATH. We also conduct evaluation on the test split of GSM8K, with experimental results shown in the Appendix A.2.

**Setup.** As introduced in §2.1, we adopt the online RFT training framework in our implementation. We use Mistral-7B (Jiang et al., 2023) as our base model. We run the SFT baseline on MATH for 3 epochs, and use the checkpoint at the first epoch as the initial checkpoint to run self-improving training. We experiment with two different types of reward functions:

- *Answer:* we just match the predicted answer with the ground-truth final answer and keep the responses with correct answers, following (Singh et al., 2023):

$$r = \mathbb{1}(\hat{a} = a^*), \tag{1}$$

- *Answer + PRM:* we train a process reward model (PRM) using the approach in Wang et al. (2024b) based on Mistral-7B. Then we combine the final answer reward and the PRM reward as:

$$r = \mathbb{1}(\hat{a} = a^*) + r_{prm}(x, \hat{y}), \tag{2}$$

where $\hat{a}, a^*$ are the predicted answer and the ground-truth answer respectively, $\mathbb{1}(\cdot)$ is the indicator function, $\hat{y}$ is the predicted solution. $r_{prm}(\cdot)$ is the PRM score. Since PRM is designed to score every step of the solution, we choose the minimal score across all steps as the score for the entire solution, following Lightman et al. (2023). We only keep responses with $r > \tau$ to train the model, and $\tau$ is a threshold. We found $\tau = 0$ is a good hyperparameter in our early trials with different thresholds, thus we keep $\tau = 0$ in all the experiments on dynamics analysis.

We train all methods for 4500 training steps. For our online RFT, we adopt an iteration interval of 500 steps, resulting in a total of 9 iterations – significantly higher than the typical 3–5 iterations reported in prior work (Singh et al., 2023). We sample 32 candidate solutions per query during training with a temperature of 1.0. We include the SFT baselines, which are trained for 3 epochs on MATH. Please see Appendix A.1 for more setup details.

**Observation 1: Exploration decreases over training, and PRM helps retain the exploration ability.** As demonstrated in Figure 3(a), the self-improvement method significantly outperforms

SFT in enhancing the model's ability to generate accurate responses through greedy decoding. However, for both methods, the performance improvements diminish over time as the number of iterations increases. From Figure 3(b), the diversity metric decreases dramatically, a similar phenomenon is observed in Wu et al. (2024). Notably, the Answer+PRM reward is able to overcome the declining trend and retain exploration. We hypothesize that filtering responses solely based on answer correctness often results in homogeneous reasoning paths, whereas the fine-grained reward strategy encourages the selection of high-quality paths, thereby preserving diversity. Examining the Pass@K-S metrics in Figure 3(c), Pass@K-S initially increases, indicating an improvement in model exploration. However, Pass@K-S subsequently declines, with Pass@K-1 accuracy falling close to the SFT baseline. The plateau pattern of exploration is not a good sign, as it suggests that the model may not learn new things to explore better. Since the reward model remains fixed during training, the model's ability stagnates once exploration halts.

**Observation 2: Exploitation keeps improving on MATH.** As shown in Figure 3(d), we observe Reward@K-S continues to improve on MATH, potentially because we are training with the MATH dataset. Since our our reward remains fixed during training, the exploitation performance is closely related to exploration of the policy model, which is a moving target as discussed in §2.2. However, as shown in our previous analysis, the model's exploration does not continually improve in all cases. We hypothesize that this is the key factor that bottlenecks self-improving training. Exploration is related to configurations such as how we sample responses from the model, how many samples to draw. Similarly, exploitation depends on how we utilize the reward to select responses. While all previous works have treated these configurations as static during training, as far as we know, the question arises: Can we adjust them dynamically to better align exploration and exploitation? We investigate this question next.

## 3   B-STaR – BALANCED SELF-TAUGHT REASONERS

In this section, we firstly introduce a metric to evaluate the interplay effect between exploration and exploitation. Next, we analyze its relationship with different configurations. Finally, we present our complete algorithm, which automatically adjusts the exploration and exploitation abilities.

### 3.1   BALANCE SCORE

§2 highlights the importance of a model's exploration and exploitation capabilities for self-improvement, emphasizing their continuous evolution during training. Now we seek a metric that could capture the interplay of these two factors. Intuitively, we hope the model can explore a diverse range of high-quality responses and with the reward function distinguishing between them. Specifically, given a query, two conditions to be satisfied for the selected responses to effectively contribute to training: (1) a substantial absolute number of high-quality responses, and (2) a high proportion of high-quality responses among the selected ones. The first condition ensures there is a sufficient quantity of high-quality data for training, while the second prioritizes maintaining a high proportion of good responses to prevent contamination of the training set with low-quality data. For example, selecting only two correct responses may result in a perfect 100% quality ratio but provide insufficient data for training. Conversely, selecting all 64 candidates – of which 16 are correct – yields a higher absolute number of high-quality responses but lowers the quality ratio to 25%, introducing too many incorrect responses into the training process. Therefore, these two capture different aspects of the data. Following these two intuitions, we propose a metric, *balance score*, to measure the balance effect – or in other words, the overall contribution of the synthetic data – given the exploration- and exploitation-related configurations. We detail it below.

For each query $x_i$, let $n_i$ represent the total number of selected responses for it, and $n'_i$ denote the number of unique, correct responses among them, then the balance score is defined as follows:

$$bs_i = \min\left(\frac{n'_i}{n^\star}, \ 1\right) \cdot \frac{n'_i}{n_i}, \tag{3}$$

where $(\frac{n'_i}{n^\star}, \ 1)$ is a discount factor that encourages the number of correct solutions to be larger than a pre-specified parameter $n^\star$ – there is no discount when we have more than $n^\star$ correct responses. We impose a cap of 1, rather than encouraging $n'_i$ to be as large as possible, because otherwise the

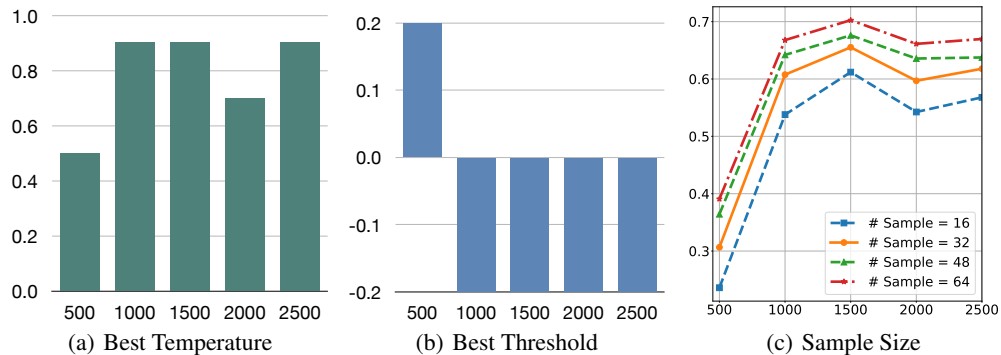

Figure 4: **Left:** the best sampling temperature to achieve the max average balance scores at different training steps; **Middle:** the best reward threshold to achieve the max average balance scores at different training steps; **Right:** the average balance scores on 600 MATH training queries at different training steps, varying the number of samples to draw per query; Every 500 steps corresponds to one iteration - for example, step 500 corresponds to the 1 st iteration, while step 1500 corresponds to the 3 rd iteration.

number of responses among queries will be severely imbalanced (Tong et al., 2024), where the easy queries will occupy most of the correct responses to maximize the average balance score. The second term, $n'_i/n_i$, is the ratio of the correct responses. We note that this ratio is always 100% if only the final-answer reward is used. $n^\star$ is the only hyperparameter in this metric, and it roughly implies the number of correct responses that we aim to have per query. Suppose we aim to select $N$ samples per iteration, and each iteration we feed in $M$ queries where $N > M$, then we simply decide $n^\star = \lceil \frac{N}{M} \rceil$. As $N$ and $M$ are just general training hyperparameters related to data loader, we never tune them in the paper and just set them as reasonable numbers as we will describe later. This means, *the balance score does not introduce additional hyperparameters for us to tune*.

Intuitively, the exploration and exploitation are desired to be controlled to maximize the average balance score – the max value of $bs_i$ is 1 and the average balance score is maximized when all the selected responses are correct and there are at least $n^\star$ correct responses for each query. Controlling exploration and exploitation is non-trivial and there could be different approaches, in this work we focus on a simple method – where we manipulate the balance through certain hyperparameter configurations such as the sampling temperature and reward threshold, which we discuss next.

### 3.2 DYNAMIC OR STATIC – ON THE CONFIGURATIONS OF EXPLORATION AND EXPLOITATION

The balance score provides a straightforward signal on how we should manipulate exploration and exploitation to be optimal. In this work, we mainly focus on adjusting some hyperparameter configurations to control exploration and exploitation. Here, we conduct a preliminary analysis to investigate whether the optimal configurations – which maximize the averaged balance score – are static or should be dynamically changing. Specifically, we obtain the policy model checkpoints and reward model checkpoints from different iterations of online RFT with the final Answer + PRM reward run, with 500 steps between each iteration. We then apply different configurations to these checkpoints, and compute the average balance scores on 600 randomly sampled MATH training queries. Below we detail the configurations we study.

**Exploration – temperature.** Sampling temperature, an important configuration influencing exploration, is first examined for its impact on balance score. With the reward threshold fixed at 0.0 and a sample size of 32, we adjust the sampling temperature to 0.5, 0.7, 0.9, and 1.1. In Figure 4(a) we show the optimal temperature we obtained that maximizes the average balance score at different iterations (training steps). In this particular setting, lower temperatures are preferred in the beginning while higher temperatures are better later on. This phenomenon can be explained by the model's shifting limitations during training. In the early stages, the model's ability to generate correct solutions is limited, so lower temperatures help sample more accurately (Yang et al., 2023). As training advances, the challenge shifts to preserving diversity in the generated solutions, requiring higher temperatures to ensure broader sampling.

| Methods | GSM 8K | | | MATH | | | APPS | | | ARC-C |
|---|---|---|---|---|---|---|---|---|---|---|
| | P@1 | P@32 | P@32-4 | P@1 | P@32 | P@32-4 | P@1 | P@32 | P@32-4 | P@1 |
| SFT | 36.6 | 88.5 | 62.2 | 17.0 | 60.8 | 31.2 | 9.3 | 43.5 | 25.5 | — |
| Rest-EM (w/o RM) | 40.5 | 89.9 | 69.8 | 22.8 | 60.0 | 33.6 | 14.5 | 43.9 | 28.2 | 70.7 |
| Rest-EM (w/ RM) | 46.3 | 90.7 | 72.2 | 24.2 | 62.8 | 37.4 | — | — | — | — |
| Iterative RFT (w/o RM) | 42.8 | 88.9 | 71.3 | 24.2 | 63.4 | 38.2 | 15.2 | 44.3 | 28.0 | 70.3 |
| Iterative RFT (w/ RM) | 46.6 | 90.2 | 74.9 | 24.4 | 62.6 | 39.0 | — | — | — | — |
| Online RFT (w/o RM) | 44.0 | 88.1 | 69.7 | 23.0 | 57.2 | 38.2 | 17.3 | 45.8 | 27.8 | 71.2 |
| Online RFT (w/ RM) | 46.8 | 91.4 | 76.5 | 23.2 | 62.6 | 39.2 | — | — | — | — |
| B-STAR | **53.8** | **93.6** | **81.0** | **27.8** | **67.2** | **42.2** | **19.6** | **49.3** | **30.7** | **73.0** |

Table 1: Comparison of self-improvement methods across MATH, GSM8K, APPS and ARC-Challenge. Methods include variants with and without a reward model ("w/ RM" and "w/o RM"). The results are based on the Mistral-7B model except for APPS that is from Llama-3-8B.

**Exploitation – threshold.** We investigate the impact of reward thresholds on balance score, which determines how the reward exploits the responses. A high reward threshold indicates that the reward model strictly exploits responses, whereas a low threshold suggests more relaxed selection criteria. In our experiments, we fix the sampling temperature at 1.0 and the sample size at 32, while varying the reward threshold and selecting responses that exceed the threshold. Figure 4(b) presents the optimal threshold that maximizes the average balance score. It indicates that a higher threshold is preferred in the beginning, but it may need to decrease as training progresses. Intuitively, this suggests that more rigorous filtering should be applied in the beginning, when the model is weaker, and the threshold should be relaxed slightly as the model becomes stronger.

### 3.3 B-STAR

Building on the findings from §3.2 that the optimal configurations to maximize balance score is dynamically changing, we propose B-STAR, short for Balanced Self-Taught Reasoners, a method that maximizes the average balance score by dynamically adjusting configurations to balance exploration and exploitation. We also include related work on dynamically adjusting configurations in the Appendix F. Specifically, we adjust temperature and threshold automatically at every iteration, to maximize the average balance score. Notably, we only need to compute the balance score on a small subset of training queries to decide the balanced configurations, thus incurring negligible additional costs compared to the baselines. For example, we only use 600 MATH queries in our experiments. Our B-STAR algorithm is illustrated in Figure 2 and summarized in Algorithm 1. There are other configurations affecting balance score, such as the number of responses drawn per query (sample size) that will influence the exploration. In our preliminary trials in Figure 4(c), larger sample size tends to be generally helpful. Therefore, we just use the maximum sample size according to our computing budget and fix it, and mainly focus on dynamic adjustments of temperature and threshold.

## 4 MAIN EXPERIMENTS

### 4.1 SETUP

We evaluate B-STAR's effectiveness in enhancing self-improvement for mathematical problem-solving, coding challenges and commonsense reasoning. Our evaluation compares several baseline methods: STaR/ResT-EM (Zelikman et al., 2022; Singh et al., 2023), Iterative RFT, and Online RFT (Shao et al., 2024). Additionally, we implement these methods with two types of reward function: "without RM" (Answer) and "with RM" (Answer+PRM), as described in §2.3. We also provide detailed baseline methods and experimental setup in Appendix B.

### 4.2 RESULTS

**Main Results.** Table 1 provides a comprehensive comparison of B-STAR with various self-improvement methods, including Rest-EM, Iterative RFT, online RFT, and their reward model variants in mathematical problem-solving, coding challenges and commonsense reasoning. The results show that B-STAR consistently achieves higher Pass@1 scores across all scenarios, highlighting its ability to effectively steer the model toward correct responses via greedy decoding. Moreover,

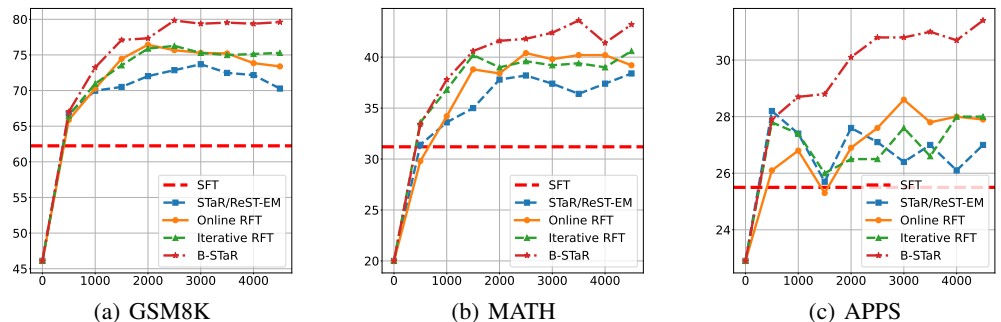

Figure 5: Pass@K-S over training steps on GSM8K, MATH and APPS. For ARC-Challenge, due to the limited response space of multiple-choice questions, Pass@K and Pass@K-S metrics (where $K > 1$) provide no additional insights and are therefore excluded.

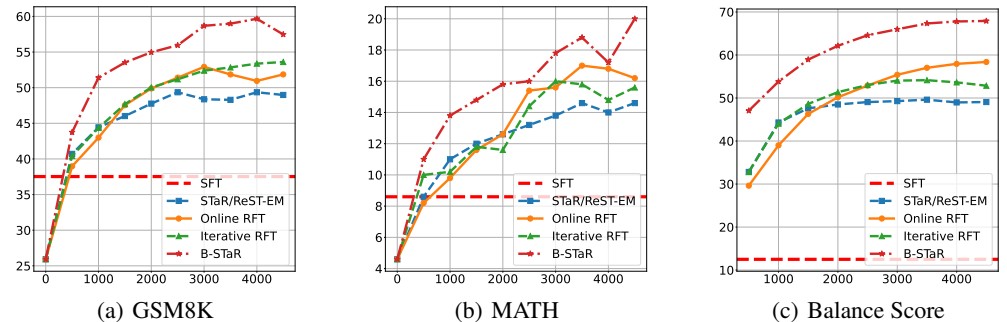

Figure 6: Reward@K-S over training steps on GSM8K (left), MATH (middle). For coding challenges and commonsense reasoning, we do not use a reward model, but instead use unit tests and ground truth answers respectively as binary rewards, so we do not report Reward@K-S. Balance Score over training steps (right).

| Step | 500 | 1000 | 1500 | 2000 | 2500 | 3000 | 3500 | 4000 | 4500 |
|---|---|---|---|---|---|---|---|---|---|
| Temperature | 0.5 | 0.8 | 0.9 | 1 | 1.1 | 1.1 | 0.9 | 1.1 | 1.1 |
| Reward threshold | 0 | -0.1 | -0.1 | -0.1 | -0.1 | -0.1 | -0.1 | -0.1 | -0.1 |
| Balance Score | 0.470 | 0.538 | 0.589 | 0.621 | 0.646 | 0.660 | 0.673 | 0.678 | 0.679 |

Table 2: Dynamic configuration adjustments by B-STAR in mathematical problem-solving. The temperature increment and reward threshold increment are both set to 0.1. Additionally, finer-grained increments for these parameters are explored in detail in Appendix D and summarized in Table 5.

B-STAR demonstrates significantly better Pass@K-S values, reflecting an enhanced exploration capacity that facilitates the generation of a wider range of high-quality responses. Notably, online RFT outperforms predominantly offline methods like Rest-EM, illustrating that dynamic, on-policy approaches strike a more effective balance between learning efficiency and performance gains.

As shown in Figure 1, all the self-improvement methods exhibit significant performance improvement after the first iteration. However, as the number of iterations increases, the growth trends of other baseline methods slows down and eventually stagnate. In contrast, B-STAR maintains a substantial growth rate and consistently outperforms other baselines. This suggests that balancing exploration and exploitation is crucial for achieving stable and efficient self-improvement.

Moreover, in Figures 5 and 6, we present the dynamic evolution of B-STAR and Online RFT throughout the self-improvement process in mathematical problem-solving. Figure 6(c) highlights B-STAR's effectiveness in balancing exploration and exploitation-related configurations, resulting in a markedly higher balance score. This optimal balance drives the consistently higher Pass@K-S and Reward@K-S scores observed in Figures 5 and 6 across both datasets. These results suggest that B-STAR not only encourages the model to generate a diverse range of accurate responses but also efficiently incorporates feedback from the reward model.

| Methods | GSM 8K | MATH |
|---|---|---|
| Online RFT | 46.8 | 23.2 |
| B-STAR (Temperature Adjustment Only) | 53.1 | 25.0 |
| B-STAR (Reward Threshold Adjustment Only) | 49.1 | 24.6 |
| B-STAR (Temperature + Reward Threshold) | **53.8** | **27.8** |

Table 3: Ablation study on dynamic adjustment in mathematical problem-solving, including temperature adjustment only and reward threshold adjustment only.

| Methods | GSM8K | MATH | APPS | ARC-C |
|---|---|---|---|---|
| SFT | 49.4 | 18.8 | 15.6 | 78.8 |
| Rest-EM (w/RM) | 60.2 | 28.2 | 16.4 | 85.5 |
| Iterative RFT (w/RM) | 55.3 | 27.2 | 17.1 | 85.2 |
| Online RFT (w/RM) | 59.7 | 27.8 | 16.9 | 85.2 |
| B-STAR | **61.6** | **29.2** | **18.1** | **86.3** |

Table 4: A comparison of self-improvement methods trained on Llama-3.1-8B across MATH, GSM8K, APPS, and ARC-Challenge, showing the highest Pass@1 results. For ARC-Challenge, we start from Llama-3.1-8B-Instruct and omit the SFT stage due to the absence of CoT data for this dataset.

**Dynamic Configuration Adjustments by B-STAR.** Table 2 presents the configurations automatically adjusted by B-STAR at various training stages, along with the resulting balance scores. Early in training, B-STAR employs a lower sampling temperature, typically around 0.5, which gradually increases as training advances. This gradual increase promotes cautious exploration during the initial phases, allowing the model to stabilize before expanding to broader sampling in later stages. Concurrently, B-STAR enforces stricter reward thresholds at the start, relaxing them as the model becomes more proficient. These strategies, as discussed in Section 3.2, enable B-STAR to maintain an effective balance between exploration and exploitation throughout the training process.

## 4.3 ABLATION STUDY AND ADDITIONAL RESULTS

**Ablation on Dynamic Adjustments.** Table 1 and Table 2 illustrate the impact of dynamically adjusting both the temperature and reward threshold on balance score and overall performance. To emphasize the importance of each configuration, we perform experiments where one configuration remains at the default settings of online RFT while the other adjusts dynamically. As shown in Table 3, dynamic adjustments to both configurations are critical, with performance significantly deteriorating when either configuration is fixed individually. To demonstrate that B-STAR's improvements stem from dynamic configuration adjustments rather than from suboptimal configuration settings, we compare B-STAR with online RFT using various fixed configuration combinations in the Appendix E. Even the best fixed configurations from grid search underperform compared to B-STaR, underscoring the crucial role of dynamic adjustments.

**Generalizing to More Powerful Models.** To assess the generalization capability of B-STAR on more powerful models, we train the model using Llama-3.1-8b and evaluate its performance across MATH, GSM8K, APPS and ARC-Challenge. Due to computational resource constraints, we set the sample size for math reasoning to 48, and the sample size for APPS and ARC-Challenge to 32, while keeping all other parameters consistent with the§4.2. The results in Table 4 show that B-STAR successfully generalizes to more powerful models.

## 5 DISCUSSION

In this paper, we reveal the importance of the balance between exploration and exploitation in self-improving training, and we propose B-STAR, a novel algorithm to strike a better balance and achieve superior performance on reasoning and coding tasks. While we adopt a simple method that manipulates exploration and exploitation through hyperparameter configurations, we expect more flexible control of these two factors can be achieved in the future to strike a better balance, such as advanced decoding approaches to directly control the exploration of the generated data, and reward model update to improve exploitation.

## ACKNOWLEDGMENT

We thank the anonymous reviewers for their feedback to help improve this paper. This work is supported by NSFC with Grant No. 62306177.

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

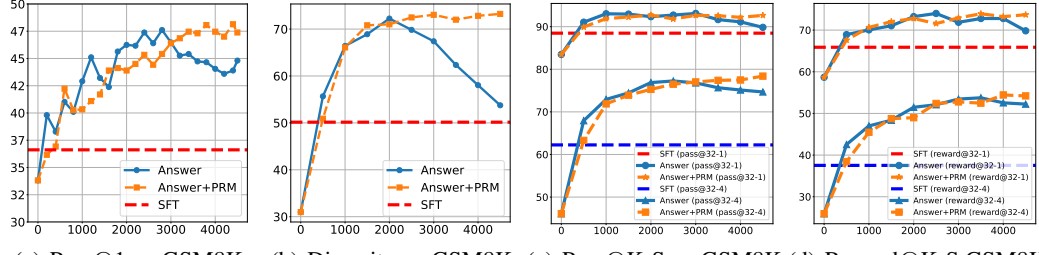

(a) Pass@1 on GSM8K    (b) Diversity on GSM8K    (c) Pass@K-S on GSM8K    (d) Reward@K-S GSM8K

Figure 7: Pass@1, Diversity, Pass@K-S and Reward@K-S over training steps on GSM8K. SFT refers to direct fine-tuning using the original dataset, "Answer" indicates matching against the ground-truth final answer, and "Answer + PRM" combines ground-truth final answer matching with PRM reward.

| Step | 500 | 1000 | 1500 | 2000 | 2500 | 3000 | 3500 | 4000 | 4500 |
|---|---|---|---|---|---|---|---|---|---|
| Temperature | 0.65 | 0.75 | 1.05 | 0.95 | 1.05 | 0.85 | 1.05 | 1.15 | 1.05 |
| Reward Thresholds | -0.02 | -0.04 | -0.09 | -0.09 | -0.14 | -0.14 | -0.14 | -0.15 | -0.06 |
| Balance Score | 0.500 | 0.557 | 0.591 | 0.626 | 0.652 | 0.665 | 0.679 | 0.682 | 0.684 |

Table 5: Finer-grained dynamic configuration adjustments by B-STAR in mathematical problem-solving.

## A  EXPERIMENTS FOR THE CASE STUDY

### A.1  EXPERIMENT SETUP

**Datasets** We use the MATH dataset for training and validate the model's mathematical reasoning ability using test sets from both the MATH (Hendrycks et al., 2021) and GSM 8K (Cobbe et al., 2021) datasets. For the MATH dataset, we follow previous settings (Lightman et al., 2023; Wang et al., 2024b; Sun et al., 2024) by using a subset of 500 representative problems (MATH500) as our test data. We uniformly sample an additional 500 problems for validation and use the remaining 4,000 problems from the MATH test set along with the original 7,500 training problems as our training data.

**Implementation details** For SFT, we use Mistral-7B (Jiang et al., 2023) as the base model with a learning rate of 5e-6, a batch size of 128, and train for 3 epochs. After the first epoch, the model (denoted as $P_0$) is used as the starting point for self-improvement. We then proceed with 9 iterations, where each iteration consists of 500 training steps with a batch size of 128. At the beginning of each iteration, we sample 32 candidate responses for each query, using a temperature of 1.0.

For the Process Reward Model (PRM), we automatically generate process annotations following the MATH-Shepherd approach (Wang et al., 2024b). Using the SFT model trained for 1 epoch, we sample 15 responses for each query in the training set. The SFT model trained for 3 epochs serves as the completer, decoding 8 solutions per step to annotate the sampled data. This process results in approximately 270 K process reward annotations. We train the reward model using the Mistral-7B base, with a learning rate of 2e-6, for 2 epochs. During rewarding, we use the lowest step score in the solution as the PRM Reward, normalize it to a range of [-1, 1] and combine it with a sparse reward to form the final reward score. We set the reward threshold to 0.0, selecting only those responses with final reward scores exceeding this threshold.

### A.2  RESULTS ON GSM8K

In Figure 7, we present the changes in Pass@1, Diversity, Pass@K-S, and Reward@K-S metrics on GSM8K. Our results indicate that incorporating PRMs leads to additional improvements on GSM8K. However, these improvements are less pronounced on MATH. We hypothesize that this is due to the difficulty of MATH problems, which may exceed the discriminative capabilities of the 7B reward model.

---

**Algorithm 1** B-STAR

---

**Require:** # Iterations $I$, initial policy $P_0$, reward model $RM$, dataset $\mathcal{D}$, temperature $t \in \mathcal{T}$, sample size $k$,
    threshold $\tau \in \Theta$, optimizer $\mathcal{O}_0$, scheduler $\mathcal{L}_0$.
**Ensure:** Final updated model $P_I$.
1: **for** $i = 1$ to $I$ **do**
2:     // Step 1: Balance Exploration & Exploitation
3:     // BS($\cdot$): the average balance score (§3.1)
4:     $t_i, \tau_i = \arg \max_{t \in \mathcal{T}, \tau \in \Theta} \mathrm{BS}(t, \tau)$
5:     // Step 2: Generate Candidates
6:     Draw $M$ queries $\{x_j\}_{j=1}^{M} \subseteq \mathcal{D}$
7:     **for** $j = 1$ to $M$ **do**
8:         **for** $m = 1$ to $k$ **do**
9:             $y_{j,m} \sim P_{i-1}(\cdot \mid x_j; t_i)$
10:         **end for**
11:     **end for**
12:     // Step 3: Evaluate via Reward Model
13:     $\mathcal{D}_i = \{(x_j, y_{j,m})\}; \quad r_{j,m} = RM(x_j, y_{j,m})$
14:     $\mathcal{D}'_i = \{(x_j, y_{j,m}) \mid r_{j,m} > \tau_i\}$
15:     // Step 4: Improve Policy Model
16:     $P_i = \mathrm{Update}(P_{i-1}, \mathcal{D}'_i, \mathcal{O}_{i-1}, \mathcal{L}_{i-1})$
17:     // Step 5: Inherit Optimizer & Scheduler
18:     $\mathcal{O}_i \leftarrow \mathcal{O}_{i-1}, \; \mathcal{L}_i \leftarrow \mathcal{L}_{i-1}$
19: **end for**

---

# B  EXPERIMENT SETUP FOR MAIN EXPERIMENTS

## B.1  BASELINE

Our evaluation compares several baseline methods: STaR/ResT-EM (Zelikman et al., 2022; Singh et al., 2023), Iterative RFT, and Online RFT (Shao et al., 2024). The STaR/ResT-EM approach involves multiple iterations, where each iteration samples from the latest policy model but resets and retrains the model from scratch. In contrast, Iterative RFT builds on each previous iteration by inheriting the checkpoint and resuming training from that point. Online RFT goes a step further by not only inheriting the checkpoint but also the optimizer state and learning rate schedule, ensuring a smoother transition across iterations.

## B.2  MATHEMATICAL PROBLEM-SOLVING

For mathematical problem-solving, we largely maintain the experimental setup from §2.3. We use the Mistral-7B model as our base and conduct SFT on the MATH training dataset for three epochs. The first epoch serves as the starting point for the model's self-improvement phase. We set the number of samples per iteration ($N$) to 67,500 and feed 11,500 MATH training queries ($M$) per iteration. We set sample size to 64 for all methods. We vary temperature from 0.5 to 1.2 in 0.1 increments and reward threshold from -1.0 to 1.0 in 0.1 increment. Additionally, finer increments for both the temperature and reward threshold are explored in Appendix D. Throughout the self-improvement process, we use Pass@1, Pass@K-S, and Reward@K-S metrics to track changes in the performance and model's exploration and exploitation capabilities.

## B.3  CODING CHALLENGES

On coding challenges, we follow Singh et al. (2023) and adopt the APPS (Hendrycks et al., 2021) dataset for both training and testing. To balance the number of responses per question, we sample 5 responses per question from the original APPS training set forming a dataset with 13K examples. We use Llama-3-8B (Dubey et al., 2024) as our base model,[3] keeping the rest of the settings consistent with those of the math domain. For baselines, we uniformly sample 32 candidate responses per query with a temperature of 0.4. For B-STAR, we explore temperatures 0.4 to 1.1 in 0.1 increment to determine the optimal configuration. We set the number of samples per iteration ($N$) to 13,500 and

---

[3]We do not use Mistral-7B as in the math domain because we found it performs poorly on coding tasks.

feed 2627 APPS training queries ($M$) per iteration. We do not apply reward models to the coding task and instead use unit tests as the binary reward, which means only the sampling temperature is automatically adjusted in B-STAR.

### B.4 COMMONSENSE REASONING

For commonsense reasoning, following Pang et al. (2024), we conduct experiments on ARC-Challenge (Clark et al., 2018), a dataset consisting of multiple-choice science questions designed to evaluate commonsense reasoning beyond mathematics and coding challenges. We start with the Mistral-7b-instruct model and omit the SFT stage due to the absence of the CoT data for this dataset. Other configurations, such as sample size and temperature, are the same as those used in the coding tasks. The ground-truth answer serves as the binary reward, and we report only Pass@1 results for the ARC-Challenge dataset. Given the constrained response space inherent to multiple-choice questions, Pass@K and Pass@K-S metrics (where $K > 1$) yield no additional insights and are therefore excluded.

## C DETAILS OF SELF-IMPROVEMENT

**Fixed Reward Function**
A fixed reward function $r(x, y)$ is a predefined, static function that does not adapt based on the training process or model parameters. For instance, binary feedback (e.g., whether a math problem is solved correctly or a unit test passes) is an example of a fixed reward function.
The fixed reward function can be represented as:

$$r(x, y) = \begin{cases} 1, & \text{if } y \text{ satisfies a predefined condition (e.g., test passes),} \\ 0, & \text{otherwise.} \end{cases} \tag{4}$$

**Reward Model**
A trained reward function $r(x, y; \phi)$ is parameterized by $\phi$ and adapts based on the training process. These reward functions (such as output-based reward models (ORMs) or process-based reward models (PRMs)) are learned from data, where the model learns to assign continuous scores based on supervision signals.
The trained reward model can be represented as:

$$r(x, y; \phi) = f(x, y; \phi) \tag{5}$$

where $f(x, y; \phi)$ is a learned function (e.g., a neural network) that maps the input $x$ and the response $y$ to a continuous reward score.

**Rejection Sampling Fine-tuning**
Rejection Sampling Fine-tuning (RFT) first samples multiple outputs from the supervised fine-tuned LLMs for each query and then trains LLMs on the sampled responses with the correct answer. Formally, the objective of RFT is to maximize the following objectives:

$$\mathcal{L}_{\text{RFT}}(\theta) = \mathbb{E}_{x \sim \mathcal{D}} \left[ \mathbb{I}(y) \log p(y \mid x; \theta) \right] \tag{6}$$

The indicator function $\mathbb{I}(y)$ is defined as:

$$\mathbb{I}(y) = \begin{cases} 1, & \text{if the answer of } y \text{ is correct,} \\ 0, & \text{if the answer of } y \text{ is incorrect.} \end{cases} \tag{7}$$

Our evaluation compares several baseline methods: STaR/ResT-EM, Iterative RFT, and Online RFT. The STaR/ResT-EM approach involves multiple iterations, where each iteration samples from the latest policy model but resets and retrains the model from scratch. In contrast, Iterative RFT builds on each previous iteration by inheriting the checkpoint and resuming training from that point. Online RFT goes a step further by not only inheriting the checkpoint but also the optimizer state and learning rate schedule, ensuring a smoother transition across iterations. Additionally, we evaluate two variants: "without RM" (Answer) and "with RM" (Answer+PRM), as described in Section § 2.3.

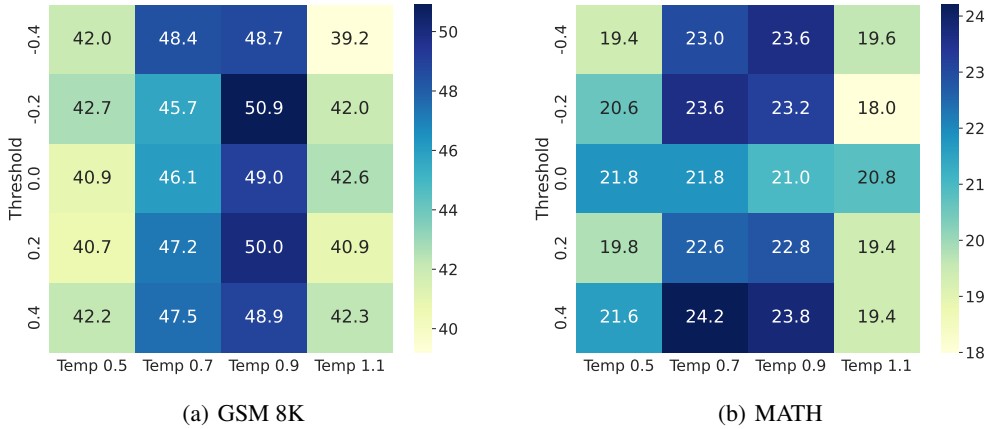

|  (a) GSM 8K | (b) MATH |

Figure 8: Performance of online RFT using different configuration combinations, where the horizontal axis represents changes in temperature and the vertical axis represents changes in reward threshold. B-STAR's GSM8K accuracy is 53.1%, while its MATH accuracy is 27.8%.

| Configuration | GSM 8K | MATH |
|---|---|---|
| Temp = 1.0; Threshold = 0.0 | 46.8 | 23.2 |
| Temp = 1.1; Threshold = -0.1 | 40.4 | 18.2 |
| B-STaR | 53.1 | 27.8 |

Table 6: Comparison of Online RFT using specific configurations and B-STaR Performance. This table reports the results with the stable hyperparameter combinations we found in our B-STaR experiments (Temperature = 1.1, Reward thresholds = -0.1) (Table 2).

## D  MORE FINE-GRAINED CONFIGURATION ADJUSTMENTS

In Section 4.1, we initially set the increments for both the temperature and reward threshold to 0.1. To explore the effects of using finer-grained increments on B-STAR, we further conduct finer-grained hyper-parameters search with the granularity of 0.05 for temperature and 0.01 for reward threshold in mathematical problem-solving setting. Table 5 illustrates how B-STAR dynamically adjusts its configuration and the resulting impact on balance score. A comparison of Table 2 and Table 5 reveals that finer-grained configuration adjustments introduce more dynamic changes to temperature and reward thresholds throughout the training process, resulting in significantly higher balance score.

## E  IMPACT OF FIXED CONFIGURATION COMBINATIONS

To confirm that the improvements achieved by B-STAR are due to its dynamic configuration adjustments rather than suboptimal configuration settings, we conduct a grid search to evaluate different configuration combinations for online RFT. The temperature values are selected from the set [0.5,0.7,0.9,1.1] and the reward thresholds are chosen from [-0.4,-0.2,0.0,0.2,0.4]. For comparison, we also include two specific configurations: the default combination from our paper, (1.0,0.0), and the parameters obtained from B-STAR's final iteration, (1.1,-0.1).

Figure 8 and Table 6 illustrate that while grid search-based configuration combinations offer some performance improvements for online RFT, they remain less effective compared to the dynamic configuration adjustments enabled by B-STAR. This further emphasizes the critical need for dynamically balancing exploration and exploitation throughout the training process.

## F    RELATED WORK OF DYNAMIC HYPERPARAMETER ADJUSTMENT

Dynamic hyperparameter optimization addresses the shortcomings of static configurations, which cannot adapt to the evolving dynamics of machine learning. Early work by Loshchilov & Hutter (2016) introduced the use of a cosine function to modulate learning rates, ensuring smoother convergence. Building on this, Baydin et al. (2017) proposed gradient-based methods to dynamically adjust learning rates in real-time by analyzing gradient trends, significantly accelerating convergence. Expanding the scope, Smith (2018) introduced a systematic approach to setting hyperparameters, such as learning rate, batch size, momentum, and weight decay, while highlighting their interdependence to improve training efficiency. Subsequently, Jaderberg et al. (2017) integrated model and hyperparameter optimization by asynchronously evolving a population of models through performance-based selection and mutation. Jomaa et al. (2019) framed hyperparameter tuning as a sequential decision-making problem, leveraging reinforcement learning to learn a policy for efficient hyperparameter selection. More recently, Baik et al. (2020) adopted a meta-learning framework to dynamically generate task- and step-specific hyperparameters, improving inner-loop optimization in few-shot learning tasks. Inspired by these innovations, our approach focuses on dynamically monitoring and balancing configurations between exploration and exploitation, optimizing the synergy between current policies and reward mechanisms to drive further performance gains.

Dynamic adjustment of hyperparameters has also proven to be crucial in pure reinforcement learning (RL) settings. For instance, Kiran & Ozyildirim (2022) highlights how adapting hyperparameters can significantly influence both the learning process and processing times in deep RL problems. Similarily, Franke et al. (2020) proposes a population-based automated RL framework that concurrently optimizes hyperparameters and neural architectures during agent training. Furthermore, Mohan et al. (2023) examines the dynamic nature of hyperparameter landscapes in RL, offering empirical evidence that these landscapes evolve over time, varying across algorithms and environments. To enhance adaptability, Bai & Cheng (2024) introduces a refined framework that emphasizes granularity and flexibility in hyperparameter adjustments, incorporating a Pairwise Learning approach to provide comprehensive guidance for improving the performance of underperforming agents.

## G    THEORETICAL JUSTIFICATION FOR EXPLORATION AND EXPLOITATION

The objective of self-improvement can be expressed in the framework of reinforcement learning as follows:

$$\pi_\theta^{t+1} = \arg \max_{\pi_\theta^t} \mathbb{E}_{x,y^* \sim \mathcal{D}, \hat{y} \sim \pi_\theta^t[\cdot|x]} \left[ R(\hat{y}, y^*) \right] \tag{8}$$

where $\mathcal{D}$ represents the dataset, $x$ and $y^*$ denote the sampled input and its corresponding ground-truth answer, respectively, and $\hat{y}$ is the sampled response. Here, $\pi_\theta^t$ corresponds to the language model in the $t$-th iteration. $R$ is the reward function. According to the policy gradient algorithm:

$$\nabla_\theta \mathbb{E}_{x,y^* \sim \mathcal{D}, \hat{y} \sim \pi_\theta^t[\cdot|x]} \left[ R(\hat{y}, y^*) \right] = \mathbb{E}_{x,y^* \sim \mathcal{D}, \hat{y} \sim \pi_\theta^t[\cdot|x]} \nabla_\theta R(\hat{y}, y^*) \log \pi_\theta^t[\hat{y}|x] \tag{9}$$

When $R(\hat{y}, y^*)$ is binary, the above equation turns to be simple data selection and supervised training loss that is exactly what we are doing. Thus, self-improvement can be viewed as a form of reinforcement learning, where maintaining a balance between exploration and exploitation is crucial and has been studied for years (Şimşek & Barto, 2006; Sutton & Barto, 2018; Weng, 2018; Wikipedia contributors, 2024). Conceptually in classic RL, exploration involves exploring the environment (analogous to reasoning tasks in our paper) by trying random actions (corresponding to sampling multiple candidates in our work) to gather more information about the environment. Exploitation, on the other hand, involves utilizing the known information to maximize the reward (similar to how we use the reward function to select data samples). Insufficient exploration may cause the training process to stagnate, while insufficient exploitation can lead to instability and large variance during training.

