# OpenReview forum: "B-STaR: Monitoring and Balancing Exploration and Exploitation in Self-Taught Reasoners"
_ICLR.cc/2025/Conference — ICLR 2025 Poster_

### Official Review · Reviewer_YTy3 · 2024-10-29

**Soundness:** 3
**Presentation:** 2
**Contribution:** 2
**Rating:** 6
**Confidence:** 3

**Summary:**

The paper is concerned with the field of self-improvement of LLMs in the context of complex reasoning tasks. Inspired by the classical RL literature, the authors identify the notions of exploration and exploitation, with their definitions adapted to the new domain, as important factors in this training procedure. They propose the B-STaR framework to address them in a dynamic way. The work first identifies factors responsible for the performance collapse observed in the self-improvement regime and then proposes to automatically adapt the training procedure to overcome it. This is done by automatically adapting two influential hyperparameters, the temperature and the reward threshold, based on the introduced query effect metric. The experimental evaluation is based on mathematical reasoning and coding tasks, showing that B-STaR leads to improved performance.

**Strengths:**

S1. The paper describes the underexplored notions of exploration and exploitation in the current research on self-improvement in complex reasoning tasks and empirically detects factors that are responsible for a balance between them. Based on that, a query effect metric is proposed to automatically adjust the related hyperparameters throughout training, which is shown to improve the performance in comparison to previous approaches.

S2. The intuitive approach to balancing the trade-off between exploration and exploitation is nicely supported with a series of experiments (Sections 2 and 3) that guide the reader to better understand the influential factors.

S3. A general overview of the approach is well summarized well with the pseudocode provided.

**Weaknesses:**

W1. My main concern about the paper is its generalizability to a broader spectrum of complex reasoning tasks. The empirical evidence is provided for two cases, one for mathematical reasoning and the other connected to coding tasks. As I am not an expert in the field, I am not aware of the number of other benchmarks that could extend the evaluation. Could the authors provide a summary of the scale of the experimental evaluation of the main papers that they compare with?

W2. The paper would strongly benefit from a more principled approach of introducing specific concepts related to self-improvement training. I would appreciate if the main or supplementary text included the definition of the SFT loss, a more detailed description of RFT, some formalization of greedy decoding, clear distinctions of trained and fixed reward functions etc. In general, formulating the paper's concepts in a mathematical language would greatly improve clarity.

W3. Some ambiguous statements are included throughout the paper which makes it difficult to follow the reasoning of the authors. For example, despite going through lines 155-160 multiple times, I was not able to grasp what the authors had in mind. Lines 167-168 mention that the distribution of candidates impacts how the reward model differentiates them, which suggests the distribution somehow influencing the reward function. I assume that the reward function is fixed and the authors meant that the distribution of candidates influences the distribution of the corresponding reward values? Or this depends on the case whether the reward is actually trained or not? I also found it hard to distinguish between the two points mentioned in lines 321-323. What is the difference between stating that `(the number of) high-quality respones among the selected ones cannot be too small' and `the percentage of the high-quality responses among the selected ones must be large enough'?I am certain that the phrasing could be improved in all of these cases.

W4. I have some concerns regarding the methodology of the empirical evaluation which is not per-se the weakness of the paper but rather of the research field. Lines 214-215 mention that, in mathematical problems, the responses are evaluated according to only the final answer and not the entire series of reasoning steps. Do I understand correctly that such evaluation would classify as correct a mathematical proof that follows wrong reasoning but arrives at some form of correct conclusion?

W5. The conclusions from Section 3.2 are based on modifying individual hyperparameters while keeping the others fixed. My concern is that the fixed values could be simply cherry-picked to fit the conclusions. Could the authors provide further motivation on why these exact fixed values should be chosen and maybe extended empirical results showing the trend for other fixed values?

W6. Tables like Table 2. should be described more extensively, along with more examples for other cases, since the adaptability of the author's approach is the main contribution of the paper. Based on Table 2. alone, it seems questionable whether any adaptation of the reward threshold parameter is necessary, since only in the beginning its value differs from other stages of the training. The increasing trend of the query effect is promising, but because only single example is provided, I am not convinced that this is always the case.

W7. Figure 6. is missing the plot for the APPS dataset. Typos: `answewrs` (line 253), `Exploration decreases over training and PRM helps retain` (line 282) <- retain what specifically?, `keeps improve` (line 295). For clarity of presentation and improved readability, the plots in Figures 5. and 6. should have the y-axis starting at zero. One could first think that plot (c) from Fig. 5 shows a higher relative improvement than plot (a) from the same figure, which does not seem to be the case after further inspection. I also strongly encourage the authors to open-source their code to allow for easy reproducibility.

**Questions:**

I have blended the questions into the Weaknesses part, hence will be happy if the authors address those extensively. My general impression is that the paper describes a nice idea of transfering the notions of exploration and exploitation into self-improvement regime, but struggles with clear communication along the way. Introducing the query effect and showing that it may be incorporated during training to adaptively modify two important hyperparameters seems to be a very interesting contribution to the field, but the empirical evidence does not clearly indicate the connection between the metric and the hyperparameters, and if it could not be simplified with just simple heuristical schedulers. Overall, the authors could elaborate more extensively on the theoretical and formal background of the paper's topic as well as the obtained results. As I am not an expert in this field, I am open to raising my rating if solid and broad arguments are provided.

---

> ### Author Response · Authors · 2024-11-23
> **Response to Reviewer YTy3 Part 1**
>
> Thanks for your time and review! We respond to your comments below.
>
> > Q1:  My main concern about the paper is its generalizability to a broader spectrum of complex reasoning tasks? Could the authors provide a summary of the scale of the experimental evaluation of the main papers that they compare with?
> >
>
> A1:  Thank you for your suggestion.  Research on self-improvement techniques for enhancing reasoning capabilities has primarily concentrated on mathematical and coding tasks, as demonstrated by works such as RFT [1], IRPO [2], REST-EM [3] and V-Star [4] . Some of these studies might further extend to commonsense reasoning tasks like ARC-Challenge and CommonsenseQA.
>
> Here, to validate the generalization of B-STaR, following IRPO [2], we add the evaluation results on the ARC Challenge dataset that consists  of multiple-choice science questions designed to evaluate commonsense reasoning beyond mathematics and coding challenges. We have added these results to Section 4 of the paper (Table 1 right and Figure 1 (d)), and we report a simplified version below as well. It demonstrates that B-STaR is not only highly effective for mathematical reasoning and code generation tasks but is also capable of generalizing to other reasoning domains. We would like to clarify that the our current evaluation on math, coding, and commonsense reasoning (for ARC) tasks aligns with or is already broader than many previous works on self-improving [1, 2, 3].
>
> |  | **ARC-Challenge (**Pass@1**)** |
> | --- | --- |
> | Base model | 65.5 |
> | Rest-EM | 70.7 |
> | Iterative RFT | 70.3 |
> | Online RFT | 71.2 |
> | **B-STAR** | **73.0** |
>
> [1] Yuan et al. Scaling Relationship on Learning Mathematical Reasoning with Large Language Models. 2023
>
> [2] Pang et al. Iterative Reasoning Preference Optimization. NeurIPS 2024.
>
> [3] Singh et al. Beyond Human Data: Scaling Self-Training for Problem-Solving with Language Models. TMLR, 2024
>
> [4] Hosseini et al. V-STaR: Training Verifiers for Self-Taught Reasoners, COLM 2024.
>
> > Q2: Specific concepts related to self-improvement training ..... formulating the paper's concepts in a mathematical language would greatly improve clarity.
> >
>
> A2: Thank you for your suggestion! We have added an introduction to specific concepts related to self-improving training and some related mathematical formulations in the Appendix B of the paper, specifically from Line 734 to Line 791.
>
> > Q3: Some ambiguous statements are included throughout the paper?
> >
>
> A3: We apologize for any confusion caused. We have improved the clarity of the mentioned contents by the reviewer. Specifically, we clarify the respective contents below.
>
> **Lines 155- 164 (previously 155-160) :**
>
> This paragraph introduces two underlying conditions essential for effective self-improvement: diverse exploration to ensure the discovery of high-quality responses and an effective reward function capable of accurately distinguishing these responses. These two conditions correspond to the concepts of exploration and exploitation in our paper.
>
> **-- "the distribution of candidates impacts how the reward model differentiates them, which suggests the distribution somehow influencing the reward function"**
>
> The distribution of candidates does not influence the reward function, and you are right that our reward function is fixed. Here we meant when the input to the fixed reward function changes, the reward function performance (the ability to distinguish the inputs) may change. This is like the same reward function could have different classification accuracy for different data distributions. We revised this part to improve the clarify (Line 166-178)
>
> **-- "What is the difference between stating that (the number of) high-quality respones among the selected ones cannot be too small and the percentage of the high-quality responses among the selected ones must be large enough"**
>
> The key difference between these two lies in that the former emphasizes the quantity of selected high-quality responses, while the latter emphasizes the ratio (but not the absolute quantity) of the high-quality responses among all selected responses. For example,  if we select only 2 responses and they are correct, then the ratio is 100% while the absolute number is only 2, which cannot give us enough training data; if we select all 64 candidates, suppose 16 of them are correct and others are wrong, then we have 16 high-quality responses, but the ratio is only 25% -- feeding many incorrect responses into training is undesirable. Therefore, these two capture different aspects of the data. We have made clarification of this to Line 319- 341.

---

> ### Author Response · Authors · 2024-11-23
> **Response to Reviewer YTy3 Part 2**
>
> > Q4: Such evaluation would classify as correct a mathematical proof that follows wrong reasoning but arrives at some form of correct conclusion?
> >
>
> A4: That’s a good point, you are right that the response would be classified as correct if the final answer is correct. Evaluating correctness solely based on the final answer can indeed cause  flawed reasoning paths to be classified as correct, and almost all the self-improvement methods admit this limitation as it is non-trivial to guarantee the correctness of the intermediate steps. One approach to mitigate this issue is to introduce finer-grained supervision signals, such as process reward models (PRMs), which evaluate reasoning step-by-step. This approach is included in our work. How to evaluate intermediate steps faithfully is still an challenging topic in this direction, which we leave as future work to explore.
>
> > Q5: The conclusions from Section 3.2 are based on modifying individual hyperparameters while keeping the others fixed..... Could the authors provide further motivation on why these exact fixed values should be chosen and maybe extended empirical results showing the trend for other fixed values?
> >
>
> A5: Section 3.2 aims to provide a preliminary analysis to motivate our methods. The purpose of Section 3.2 is mainly to show the optimal hyperparameters for the best query effect may not be constant over training, thus we only provide a case where they are not constant, and those fixed values are commonly used heuristic values without cherry-picking (temperature=1, reward threshold=0, sample size=32).  While we analyze sample size, temperature, reward threshold in Section 3.2, we did not really use any conclusion on temperature and reward threshold in our B-STaR algorithm directly -- in the actual B-STaR algorithm, we perform grid search of temperature and threshold together to select the best hyperparameters,  rather than fixing the other to select each one separately. We did use the conclusion on sample size in our final algorithm, where we always chose the sample size to be the max allowable value within our budget. We agree that this conclusion on sample size is not comprehensively validated, this is because our B-STaR is designed to mainly focus on adjusting temperature and reward threshold in the current version, and we intended to fix the sample size and not adjust it dynamically to save the adjustment cost. We will explore adjusting sample size together with the other hyperparameters in the future.
>
> > Q6: Whether any adaptation of the reward threshold parameter is necessary, since only in the beginning its value differs from other stages of the training？
> >
>
> A6: To answer this question, we conduct two new experiments  to demonstrate the necessity of dynamic adjustment of reward threshold.
>
> 1. We perform an ablation study where we only adjust the temperature while fixing the reward threshold. Specifically, we fix the reward threshold as 0.0 that is the one used by the default online RFT setting. The results are shown below.
>
>
>     |  | MATH | GSM 8K |
>     | --- | --- | --- |
>     | Online RFT | 23.2 | 46.8 |
>     | B-STaR (only adjust temperature) | 25.0 | 53.1 |
>     | B-STaR (adjust temperature and reward threshold) | **27.8** | **53.8** |
>
>     The results demonstrate that changes in the reward threshold are helpful,  and fixing the reward threshold leads to a noticeable drop.
>
> 2. The finer-grained hyper-parameter adjustments in B-STaR (Appendix C):  In the previous experiments, due to the coarse adjustment with the granularity of 0.1, the dynamic change in hyperparameter was not obvious. Therefore, we further conduct finer-grained hyper-parameters search with the granularity of 0.05 for temperature and 0.01 for reward threshold. In Table 3, we observe more dynamic change to temperature and reward thresholds throughout the training process.
>
> Therefore, we conclude that the dynamic tuning is necessary for the sustained growth and higher performance of B-STaR.

---

> > ### Author Response · Authors · 2024-11-23
> > **Response to Reviewer YTy3 Part 3**
> >
> > > Q7: The increasing trend of the query effect is promising, but because only single example is provided, I am not convinced that this is always the case？
> > >
> >
> > A7: Thank you for your suggestion.
> >
> > We incorporate the query effect trend for the ARC Challenge task, summarized in the table below:
> >
> > | **Query Effect** | **100** | **200** | **300** | **400** | **500** | **600** | **700** | **800** | **900** |
> > | --- | --- | --- | --- | --- | --- | --- | --- | --- | --- |
> > | Online RFT | 0.939 | 0.966 | 0.880 | 0.572 | 0.243 | 0.107 | 0.064 | 0.052 | 0.048 |
> > | B-STaR | 0.947 | 0.973 | 0.980 | 0.991 | 0.993 | 0.992 | 0.995 | 0.994 | 0.993 |
> >
> > The results demonstrate that B-STaR achieves an increasing trend of the query effect on ARC Challenge as well.
> >
> > > Q8: Figure 6 is missing the plot for the APPS dataset？
> > >
> >
> > A8: As mentioned in our setup (Line 463-Line 465), we did not use a reward model for the coding task, but instead used unit tests as a binary reward, so we did not report the Reward@k-s for APPS.
> >
> > > Q9: Typos
> > >
> >
> > A9: Thank you for pointing out our typos! We have corrected them in the updated PDF.

---

> > > ### Comment · Reviewer_YTy3 · 2024-11-25
> > >
> > > Thank you for an exhaustive response to my comments. I went through it and the updated version of the paper, and regard the changes as an overall improvement to the paper's quality. I have some additional comments below, which I would be happy to hear about from the authors before making my final decision.
> > >
> > > **Lines 154-164**. While the two important factors are motivated heuristically, they are not grounded theoretically. Developing at least some conceptualization of why these are important would further improve the paper. I am not deeply involved in the RL literature, but considering that the authors find some connection of their approach to this domain, it would be great to also ground the work using some theoretical framework, as is often the case with works in RL. This is also connected to the next comment.
> > >
> > > **Lines 319-341**. Why do we strictly regard wrong responses as those that are of lower quality? Intuitively, if the responses are wrong and the reward model correctly identifies them as such, the resulting signal is correct to use for further training. To be clear, I am not saying that the authors are wrong here. I just wonder what is the justification for this reasoning.
> > >
> > > **Typos**. I may be a bit meticulous here, but two of the typos mentioned previously are still in the text.

---

> ### Author Response · Authors · 2024-11-26
> **Response to Reviewer YTy3 Part 4**
>
> Thank you for the response!  For your additional comments, we address them below.
>
> > Q1：While the two important factors are motivated heuristically, they are not grounded theoretically. Developing at least some conceptualization of why these are important would further improve the paper.
>
> A1: Thank you for this suggestion. The objective of self-improvement can be expressed in the framework of reinforcement learning as follows:
>
> $$
> \pi_\theta^{t+1} = \arg\max_{\pi_\theta^t} E_{x,y^* \sim \mathcal{D}, \hat{y} \sim \pi_\theta^t[\cdot|x]} \left[ R( \hat{y}, y^*) \right]
> $$
>
> where $\mathcal{D}$ represents the dataset, $x$ and $y^*$  denote the sampled input and its corresponding ground-truth answer, respectively, and $\hat{y}$ is the sampled response. Here,  $\pi_\theta^t$  corresponds to the language model in the $t$-th iteration. $R$ is the reward function. According to the policy gradient algorithm,
>
> $$
> \nabla_{\theta} E_{x,y^* \sim \mathcal{D}, \hat{y} \sim \pi_\theta^t[\cdot|x]} \left[ R( \hat{y}, y^*) \right] = E_{x,y^* \sim \mathcal{D}, \hat{y} \sim \pi_\theta^t[\cdot|x]}\nabla_{\theta}R(\hat{y}, y^*)\log \pi_\theta^t[\hat{y}|x]
> $$
>
> When $R(\hat{y}, y^*)$ is binary, the above equation turns to be simple data selection and supervised training loss that is exactly what we are doing. Thus, self-improvement can be viewed as a  form of reinforcement learning, where maintaining a balance between exploration and exploitation is crucial  and has been studied for years  [1,2,3,4]. Conceptually in classic RL, exploration involves exploring the environment (analogous to reasoning tasks in our paper) by trying random actions (corresponding to sampling multiple candidates in our work) to gather more information about the environment. Exploitation, on the other hand, involves utilizing the known information to maximize the reward (similar to how we use the reward function to select data samples). Insufficient exploration may cause the training process to stagnate, while insufficient exploitation can lead to instability and large variance during training. We appreciate the reviewer's suggestion, and we have added this discussion to Appendix F due to the limited time and space available during the rebuttal period. For the next formal revision, we will ensure to clarify the connection above in the main body of the paper as well.
>
> [1] Wikipedia contributors. "Exploration-exploitation dilemma." *Wikipedia, The Free Encyclopedia*, 25 Sept. 2024, 07:07 UTC. Available at: [https://en.wikipedia.org/w/index.php?title=Exploration-exploitation_dilemma&oldid=1247645791](https://en.wikipedia.org/w/index.php?title=Exploration-exploitation_dilemma&oldid=1247645791). Accessed 26 Nov. 2024
>
> [2] Weng, L. (2018). *The multi-armed bandit problem and its solutions*. Retrieved from https://lilianweng.github.io
>
> [3] Şimşek et al. An intrinsic reward mechanism for efficient exploration. ICML 2006
>
> [4] Sutton et al. Reinforcement learning: An introduction. MIT press, 2018.
>
> > Q2: Why do we strictly regard wrong responses as those that are of lower quality?
>
> A2:   That’s a good point. In previous works, wrong responses were often regarded as those of lower quality when additional reward models were not used. In such cases, the correctness of the responses is typically the only signal available for selecting data, as practiced in most works [1, 2, 3]. In our paper, we follow this approach for the APPS and ARC-Challenge datasets, where we do not train a reward model. However, for mathematical reasoning, where we did train additional reward models, we combined the correctness of the response and the scores from the reward models to compute the final reward (as described in Eq. 1 of the paper). Therefore, in our case, **we did not strictly regard wrong responses as lower quality. Instead, responses with incorrect answers but high scores from the reward model were still likely to be selected in our approach.**
>
> [1] Avi et al. Beyond human data: Scaling self-training for problem-solving with language models. TMLR, 2024
>
> [2] Yuan et al. Scaling Relationship on Learning Mathematical Reasoning with Large Language Models. 2023
>
> [3] Zelikman et al. STaR: Bootstrapping Reasoning With Reasoning. NeurIPS 2022

---

> > ### Author Response · Authors · 2024-11-26
> > **Response to Reviewer YTy3 Part 5**
> >
> > > Q3: two of the typos mentioned previously are still in the text?
> >
> > A3: We are sorry for missing the previous typos, and thank you for helping us catch them! we have fixed them in the pdf now.  Particularly for the clarity issue on "Exploration decreases over training and PRM helps retain", we have changed it to be"Exploration decreases over training, and PRM helps retain the exploration ability"
> >
> > > For clarity of presentation and improved readability, the plots in Figures 5. and 6. should have the y-axis starting at zero. I also strongly encourage the authors to open-source their code to allow for easy reproducibility.
> >
> > Regarding the suggestion to start the y-axis at zero in Figures 5 and 6, we would like to clarify that the models exhibit different performance ranges on these benchmarks, making a direct comparison between the figures infeasible. This presentation style aligns with several well-known works, such as GPT-2’s Figure 1 [1], CoT’s Figure 6 [2], and Llama’s Figure 2 [3].  Additionally, we confirm that we will open-source our code after the review period.
> >
> > [1] Radford et al. Language models are unsupervised multitask learners. OpenAI blog, 2019
> >
> > [2] Wei et al. Chain-of-thought prompting elicits reasoning in large language models. NeurIPS 2022
> >
> > [3] Touvronet al. Llama: Open and efficient foundation language models. Preprint 2023

---

> > > ### Comment · Reviewer_YTy3 · 2024-11-26
> > >
> > > Thank you for addressing my comments. After carefully reading the authors responses to other reviewers concerns, and considering that mine were well-addressed, I decided to update my score from 5 to 6. Below, I include some additional remarks to improve the next revisions of the paper.
> > >
> > > I really appreciate framing the problem in the RL context mathematically, even despite its simplicity. This should give the reader a clearer introduction to the problem. Moreover, it adds additional value to the work, as the connection between RL and self-improvement is now evident and, as the authors mention, the balance of exploration and exploitation is needed in that domain. I encourage the authors to update the next version with this formulation as precisely as possible. To follow on that and other reviews, the current approach lacks theoretical justifications, which I would regard as my main reason for not giving an even higher score. In my honest opinion, the beauty and value of some earlier groundbreaking works in RL stems not only from good ideas but also precise theoretical framing. Here, [Soft Actor-Critic](https://arxiv.org/abs/1801.01290) is my favorite example. While the idea is simple and based on additional regulizarion with entropy, the paper delves more deeply into the theoretical matters of this trick. And it also started a broader [discussion](https://proceedings.mlr.press/v97/ahmed19a/ahmed19a.pdf) on the topic of including entropy in the objective. I would look for improvements there, i.e., maybe the query effect metric could be somehow derived (even in a simplified scenario) by quantifying the notion of exploration-exploitation balance through some objective that was not considered by previous works on self-improvement and is maybe connected to entropy. At this moment, the proposed solution is effective in adapting the hyperparameters, but while it partially addresses the problem of plateauing or collapse, it opens new problems connected to *why* this adaptation actually helps. Overall, the paper would additionally benefit from clearer writing, but this means different things for different people and largely depends on personal preference.

---

> > > > ### Author Response · Authors · 2024-11-27
> > > >
> > > > Thank you for your effort and valuable suggestions during the review process！We will provide more rigorous mathematical formulations to introduce the problem, strengthen the theoretical foundations of our approach, and enhance the overall writing quality in the next version!

---

### Official Review · Reviewer_hcTF · 2024-11-01

**Soundness:** 3
**Presentation:** 2
**Contribution:** 3
**Rating:** 6
**Confidence:** 4

**Summary:**

This paper introduces B-STAR, a framework aimed at improving the self-improvement process of large language models in complex reasoning tasks, such as mathematical problem-solving and coding. The primary challenge addressed is the difficulty in balancing exploration (the model's ability to generate diverse, high-quality responses) and exploitation (the model's ability to effectively select the best responses) in iterative training processes without extensive human-labeled data.

**Strengths:**

**1.Identification of Balance Problem and Innovative Framework.** The introduction of dynamic parameter adjustment based on the "query effect" metric is an innovative approach to optimize performance across training iterations.

**2.Clear Explanation of Exploration and Exploitation Dynamics**: By quantifying exploration and exploitation using metrics such as Pass@K and Reward@K-S, the paper provides insights into the dynamics of these factors, which are often overlooked in self-improvement research. This deepens understanding of the critical factors for model self-improvement.

**Weaknesses:**

**Limited Applicability Beyond Specific Tasks**: The paper focuses primarily on mathematical reasoning and code generation tasks, which limits the perceived applicability of B-STAR to other domains (e.g., open-domain question answering, language understanding) that query effect is hard to evaluate. And since QE is critical for following dynamical adjustment, this greatly limits the application.

**Lack of Theoretical Foundation for Query Effect Metric**: Although the query effect metric is practical for optimizing exploration and exploitation, the paper lacks a rigorous theoretical justification for this metric. This could raise questions about its generalizability and effectiveness across a broader range of tasks and model architectures.

**Unkown advantage over hyperparameter shift.** It seems that the final dynamics of the parameters do not fluctuate a lot. It should the excluded that most of the benefit is gain by adding an offset rather that dynamically change these parameters.

**Questions:**

**1.Ablation and Related Work**: From figure 4, it seems that after 1000 iterations, parameters like temperature and threshold remain constant. So essentially the dynamics freeze. *Does this imply that original hyperparameter is simply not set to optimal?* It is important to see how static shift of hyperparameter can help to fully verify the effectiveness and necessity of the proposed method. Also, there lacks an explicit discussion and reference to the line of work that dynamically change the hyperparameter.

**2.Query Effect Metric Validity**: How robust is the query effect metric across different tasks beyond mathematical reasoning and code generation? Without a theoretical foundation, is there a risk that this metric might not generalize well to other domains that require different types of exploration-exploitation balances?

**3.Reward Model Dependency**: The paper relies heavily on reward models (both answer-based and process-based). How does the performance of B-STAR change if the reward model is not well-aligned with task objectives? In cases where defining an accurate reward model is challenging, can B-STAR still maintain its effectiveness?

---

> ### Author Response · Authors · 2024-11-23
> **Response to Reviewer hcTF Part 1**
>
> Thanks for your time and comments! We respond to your comments below.
>
> > Q1: Limited Applicability Beyond Specific Tasks:  Evaluation on other reasoning tasks, the robust of the query effect metric across different tasks beyond mathematical reasoning and code generation？
> >
>
> A1: Thank you for your suggestion. We follow IRPO [3] and add the evaluation results on the ARC Challenge dataset, a benchmark consisting of multiple-choice science questions designed to evaluate commonsense reasoning beyond mathematics and coding challenges, as shown below.
>
> |  | ARC-Challenge (Pass@1) |
> | --- | --- |
> | Base model | 65.5 |
> | Rest-EM | 70.7 |
> | Iterative RFT | 70.3 |
> | Online RFT | 71.2 |
> | **B-STAR** | **73.0** |
>
> The results clearly indicate that B-STaR outperforms other baseline methods, achieving a significant improvement in reasoning performance. We note that the our current evaluation on math, coding, and commonsense reasoning (for ARC) tasks aligns with or is already broader than many previous works on self-improving [1, 2, 3]. These added results and more details are also included in Section 4 of the paper.
>
> For other tasks such as open-domain QA and language understanding suggested by the reviewer, CoT typically does not help these tasks [4]. Also, these tasks are typically not considered as "reasoning" tasks and rarely used  for self-improving reasoning in the literature. As our paper specifically focuses on complex reasoning, these non-reasoning tasks  fall outside the scope of our paper.
>
> [1] Avi et al. Beyond human data: Scaling self-training for problem-solving with language models. TMLR, 2024
>
> [2] Zhang et al. Small Language Models Need Strong Verifiers to Self-Correct Reasoning. COLM 2024.
>
> [3]  Pang et al. Iterative Reasoning Preference Optimization. NeurIPS 2024.
>
> [4] Sprague et al. To CoT or not to CoT? Chain-of-thought helps mainly on math and symbolic reasoning. Preprint 2024.
>
> > Q2: Unknown advantage over hyperparameter shift? It seems that the final dynamics of the parameters do not fluctuate a lot. It should the excluded that most of the benefit is gain by adding an offset rather that dynamically change these parameters.
> >
>
> A2:  This is a good point. To answer this question, we conduct two new experiments to demonstrate the necessity of dynamic adjustment of hyperparameters.
>
> 1. First, we report comprehensive grid search results of various temperature and reward threshold combinations, where these hyperparameters are decided in the beginning and remain static. We present the complete results of online RFT in Figure 7 and Table 4 of Appendix D. Below we briefly show four optimal configurations found in grid search and compare them with our B-STaR.
>
> | **Configuration** | **GSM 8K** | **MATH** |
> | --- | --- | --- |
> | Temp=0.9; Threshold=-0.2 | 50.9 | 23.3 |
> | Temp=0.7; Threshold=0.4 | 47.5 | 24.2 |
> | Temp=1.0; Threshold=0.0 | 46.8 | 23.2 |
> | Temp=1.1; Threshold=-0.1 | 40.4 | 18.2 |
> | **B-STaR** | **53.1** | **27.8** |
>
> The results indicate that while fixed hyperparameter values derived from grid search can lead to some performance improvements in online RFT, they remain inferior to the adaptive parameter adjustments offered by B-STaR. This highlights the necessity of dynamic parameter optimization in achieving superior performance. These results and more details are added as Appendix D of the updated PDF.
>
> 2. Second, part of the reason that previous experiments did not show significant fluctuations might be that  we adopted a coarse adjustment scheme with the granularity of 0.1. To verify this, we further conduct finer-grained hyper-parameters search with the granularity of 0.05 for temperature and 0.01 for reward threshold. In Table 3 of Appendix C, we observe more dynamic changes throughout the training process.
>
> > Q3: From figure 4, it seems that after 1000 iterations, parameters like temperature and threshold remain constant. So essentially the dynamics freeze. Does this imply that original hyperparameter is simply not set to optimal?
> >
>
> A3: Please refer to A2 to Q2, which verifies the effectiveness and the necessity of the proposed B-STaR method.

---

> > ### Author Response · Authors · 2024-11-23
> > **Response to Reviewer hcTF Part 2**
> >
> > > Q4: An explicit discussion and reference to the line of work that dynamically change the hyperparameter?
> > >
> >
> > A4: Thanks for the suggestion! Dynamic hyperparameter optimization is common in many other machine learning scenarios, for instance, [1] employs a cosine function to modulate learning rates, ensuring smoother convergence and enhanced performance. Building on this, [2] proposed gradient-based methods to dynamically adjust learning rates in real-time by analyzing gradient trends, significantly accelerating convergence. We have added a dedicated discussion of this line of works in Appendix E of the updated PDF. However, we note that we are the first to study hyperparameter dynamic changing in self-improving LLM reasoning as far as we know.
> >
> > [1] Loshchilov et al. SGDR: Stochastic Gradient Descent with Warm Restarts. 2016
> >
> > [2] Baydin et al. Online Learning Rate Adaptation with Hypergradient Descent. 2017
> >
> > > Q5:  How does the performance of B-STAR change if the reward model is not well-aligned with task objectives?
> > >
> >
> > A5: This is an interesting and important question. Indeed,  a well-aligned reward model is essential for self-improvement, serving as a foundational prerequisite for most self-improving methods in general . Therefore, we think it is necessary for the reward model to be effective for the data at hand, and how to obtain generalizable reward models that can effectively discriminate good responses for many tasks remain as a central yet challenging research problem in self-improving. In cases where defining an accurate reward model is challenging, reward models can be defined to mimic (subjective) human preferences, just as previous works practiced for self-improving non-reasoning tasks [1, 2, 3].
> >
> > [1] Yuan et al. Self-rewarding language models. ICML 2024.
> >
> > [2] Wu et al. Meta-rewarding language models: Self-improving alignment with llm-as-a-meta-judge. 2024.
> >
> > [3] Gulcehre et al. Reinforced self-training (rest) for language modeling. 2024.

---

> > > ### Comment · Reviewer_hcTF · 2024-11-27
> > >
> > > Thank you for your response. My concern are mostly resolved. Hope to see these new results in revision. Also, since you also agree that dynamic hyperparameter optimization is common in many other machine learning scenarios, I still suggest to add adequate reference to the work in RL domain that also deals with dynamically change the hyperparameters, but maybe in pure RL setting. I raise my score to 6.

---

> > > > ### Author Response · Authors · 2024-11-27
> > > >
> > > > Thank you for your valuable feedback and suggestions during the review process！We will incorporate additional references regarding dynamic hyperparameter optimization in the RL domain in our next version!

---

### Official Review · Reviewer_Gb3F · 2024-11-03

**Soundness:** 3
**Presentation:** 3
**Contribution:** 3
**Rating:** 6
**Confidence:** 3

**Summary:**

The paper is addressing how to improve self-taught reasoners. It addresses the challenges of iterative self-improvement for reasoning tasks in AI models, particularly in mathematical problem-solving. With limited access to high-quality, human-annotated data, models are trained to refine themselves via generated outputs. However, performance gains often stagnate after a few iterations, with degradation in exploratory diversity and effectiveness of rewards. To counter these issues, B-STAR introduces a dynamic framework that monitors and balances exploration (generating high-quality candidate responses) and exploitation (selecting optimal responses using reward mechanisms).

Key innovations include metrics like "Pass@K" for exploration and "Best-of-K" for exploitation, which quantify the model's ability to generate diverse and correct outputs while ensuring selected outputs are useful for training. B-STAR adapts configurations such as sampling temperature and reward thresholds, adjusting them based on "query effect" scores to optimize exploration-exploitation balance iteratively. In each iteration, the algorithm dynamically adjusts configurations, specifically sampling temperature and reward thresholds, to maximize the "query effect" score, which ensures an effective balance. The model then generates multiple responses per query, selecting only those with high rewards to use as training data for the next iteration. This process is repeated across iterations, inheriting the optimizer and learning rate to maintain consistency in online training and continually improve model performance.

The experiments in B-STAR evaluate its effectiveness on mathematical reasoning and coding tasks, comparing it against other self-improvement methods like Iterative RFT and Online RFT. B-STAR consistently outperforms these baselines in generating accurate responses, as shown by higher Pass@1 and Pass@K-S scores, which indicate both accuracy and diversity in responses. In mathematical tasks, B-STAR maintains exploration across iterations, unlike other methods where diversity declines, highlighting its balanced approach to exploration and exploitation. For coding challenges, B-STAR achieves steady improvement across iterations, suggesting that dynamic configuration adjustments improve response quality over time. The experiments validate B-STAR’s approach in sustaining a higher growth rate in performance by adaptively balancing exploration and exploitation.

**Strengths:**

- Dynamic Balancing of Exploration and Exploitation
- Comprehensive Metrics for Monitoring
- Empirical Validation with strong results
- Insightful Analysis of Self-Improvement Dynamics

**Weaknesses:**

- Reliance on fixed reward models
- complexity of configuration tuning
- limited generalization across task types
- potenitalluy saturate in long iterative training

**Questions:**

- address weaknesses

---

> ### Author Response · Authors · 2024-11-23
> **Response to Reviewer Gb3F**
>
> Thanks for your time and encouraging review! We respond to your comments below.
>
> > Q1: Reliance on fixed reward models
> >
>
> A1: We use fixed reward models following most literature in this direction [1, 2, 3]. We agree that it is possible to update the reward models during training to obtain further gains, we leave exploration of this for future work.
>
> [1] Hosseini et al. V-STaR: Training Verifiers for Self-Taught Reasoners, COLM 2024
>
> [2] Sun et al. Easy-to-Hard Generalization: Scalable Alignment Beyond Human Supervision, 2024
>
> [3] Wang et al. Math-Shepherd: Verify and Reinforce LLMs Step-by-step without Human Annotations, ACL 2024
>
>
> > Q2: complexity of configuration tuning
> >
>
> A2:  We clarify that our method does not increase the complexity of configuration tuning, instead, it simplifies and reduces the efforts to tune hyperparameters such as temperature and reward threshold. This is because the proposed B-STaR automatically decides the critical hyperparameters through the introduced Query Effect metric, with negligible cost during training (Line 420-422). In contrast, previous methods need to tune these hyperparameters manually after going through multiple expensive training processes in a trial-and-error fashion.
>
>
> > Q3: limited generalization across task types
> >
>
> A3: Thank you for your suggestion. We follow IRPO [3] and add the evaluation results on the ARC Challenge dataset, a benchmark consisting of multiple-choice science questions designed to evaluate commonsense reasoning beyond mathematics and coding challenges, as shown below.
>
> |  | ARC-Challenge (Pass@1) |
> | --- | --- |
> | Base model | 65.5 |
> | Rest-EM | 70.7 |
> | Iterative RFT | 70.3 |
> | Online RFT | 71.2 |
> | **B-STAR** | **73.0** |
>
> The results clearly indicate that B-STaR outperforms other baseline methods, achieving a significant improvement on ARC Challenge as well. This demonstrates that B-STaR is not only highly effective for mathematical reasoning and code generation tasks but is also capable of generalizing to other complex reasoning domains. We note that the our current evaluation on math, coding, and commonsense reasoning (for ARC) tasks aligns with or is already broader than most previous works on self-improving [1, 2, 3]. These added results and more details are also included in Section 4 and Figure 1 (d) of the paper.
>
> [1] Avi et al. Beyond human data: Scaling self-training for problem-solving with language models. TMLR, 2024
>
> [2] Zhang et al. Small Language Models Need Strong Verifiers to Self-Correct Reasoning. COLM 2024.
>
> [3]  Pang et al. Iterative Reasoning Preference Optimization. NeurIPS 2024.
>
> > Q4: potentially saturate in long iterative training
> >
>
> A4: This is a good point. We agree that with extended iterative training, model performance will eventually saturate due to many factors such as the training dynamics, the model's intrinsic capabilities, the data coverage, and the quality of the reward function. While overcoming all these fundamental limitations remain challenging, our proposed approach, B-STAR, specifically focuses on the training dynamics aspect and achieves more sustained growth and higher performance compared to baseline methods. Achieving scalable and sustainable improvements over long iterative training requires to address many other challenges, which we leave as future work to explore.

---

> > ### Author Response · Authors · 2024-12-02
> > **A Kind Reminder for Reading the Response**
> >
> > Dear Reviewer Gb3F,
> >
> > Thank you for your insightful suggestions.
> >
> > In response to your feedback, we have revised the paper and added experiments to demonstrate B-STaR's generalization to other reasoning tasks. We have also addressed your concerns regarding our use of a fixed reward model, clarified how B-STaR maintains low complexity, and explained how it achieves more sustained growth during long iterative training compared to alternative methods. As the rebuttal period is nearing its end, we would greatly appreciate if you could review our response to confirm whether it adequately addresses your concerns.
> >
> > Thank you for your time and consideration,
> >
> > The authors

---

### Author Response · Authors · 2024-11-23
**General Response to Reviewers and Revision Submitted**

We thank all the reviewers for their insightful comments and suggestions! We have revised the PDF to reflect the reviewers' comments (the revisions are marked with blue text in the pdf), and responded to each reviewer separately in the respective thread. Here we summarize the main revisions of the manuscript.

1. We evaluated B-STaR on ARC-Challenge, another commonly used reasoning dataset consisting of multiple-choice science questions outside math and coding (Section 4, Reviewer Gb3F, hcTF and YTy3).
2. We added finer-grained hyperparameter adjustments during B-STaR training, revealing the dynamic changes in hyperparameters including temperature and reward threshold (Appendix C, Reviewer hcTF and YTy3).
3. We performed baseline experiments for grid-search sets of static hyperparameter combinations, highlighting the necessity of dynamic adjustment (Appendix D, Reviewers hcTF and YTy3).
4. We introduced specific concepts related to self-improving training to enhance clarify, including detailed descriptions of SFT, RFT, greedy decoding, and clear distinctions between trained and fixed reward functions, etc. (Appendix B, Reviewer YTy3).

---

### Author Response · Authors · 2024-11-25
**A Kind Reminder for Reading the Response**

Dear Reviewers,

We have revised the paper and added many additional results to address your comments. Since the rebuttal period is closing very soon, can you please check the response to see whether it mitigates your concerns? We would greatly appreciate that!

Thank you,

The authors

---

### Meta-Review · Area_Chair_i5hR · 2024-12-21

**Metareview:**

The paper proposes B-STaR, a novel framework for dynamically balancing exploration and exploitation in self-improving large language models. The proposed framework is well-motivated, and the empirical results, particularly the new evaluations on the ARC Challenge provided during the rebuttal, demonstrate clear improvements over baseline methods. Overall, I share the reviewers’ positive views and believe the paper makes a valuable contribution to the literature.

Meanwhile, shared concerns remain regarding the lack of theoretical grounding for the query effect metric, limited generalizability to broader reasoning tasks, and some clarity issues in the presentation. While the rebuttal addressed several of these points effectively, the authors are strongly encouraged to incorporate the reviewers’ feedback, particularly to strengthen the theoretical justification and improve clarity, in future revisions.

**Additional Comments On Reviewer Discussion:**

During the rebuttal, reviewers raised concerns about the generalizability of the framework beyond reasoning tasks, lack of theoretical grounding for the query effect metric, and some clarity issues in the presentation. The authors addressed these by adding ARC Challenge results to demonstrate broader applicability, providing conceptual links to reinforcement learning to partially justify the query effect, and clarifying ambiguities while committing to further improvements. New experiments also validated the necessity of dynamic hyperparameter tuning.

---

### Decision · Program_Chairs · 2025-01-22

Accept (Poster)